# Cell elimination strategies upon identity switch via modulation of *apterous* in *Drosophila* wing disc

Olga Klipa[1,2], Fisun Hamaratoglu[1,2]*

**1** Center for Integrative Genomics, University of Lausanne, Lausanne, Switzerland, **2** School of Biosciences, Cardiff University, Cardiff, United Kingdom

* hamaratoglu@cardiff.ac.uk

**Data Availability Statement:** All relevant data are within the manuscript and its Supporting Information files.

**Funding:** This work has been supported by the Swiss National Science Foundation professorship

## Abstract

The ability to establish spatial organization is an essential feature of any developing tissue and is achieved through well-defined rules of cell-cell communication. Maintenance of this organization requires elimination of cells with inappropriate positional identity, a poorly understood phenomenon. Here we studied mechanisms regulating cell elimination in the context of a growing tissue, the *Drosophila* wing disc and its dorsal determinant Apterous. Systematic analysis of *apterous* mutant clones along with their twin spots shows that they are eliminated from the dorsal compartment via three different mechanisms: relocation to the ventral compartment, basal extrusion, and death, depending on the position of the clone in the wing disc. We find that basal extrusion is the main elimination mechanism in the hinge, whereas apoptosis dominates in the pouch and in the notum. In the absence of apoptosis, extrusion takes over to ensure clearance in all regions. Notably, clones in the hinge grow larger than those in the pouch, emphasizing spatial differences. Mechanistically, we find that limiting cell division within the clones does not prevent their extrusion. Indeed, even clones of one or two cells can be extruded basally, demonstrating that the clone size is not the main determinant of the elimination mechanism to be used. Overall, we revealed three elimination mechanisms and their spatial biases for preserving pattern in a growing organ.

## Author summary

As development proceeds, cells become more specialized and the compartmentalization ensures spatial separation of the specialized cells. This process of pattern formation is rather well understood. How the pattern is maintained afterwards though is largely unknown. Using the *Drosophila* wing disc as a model organ, we examined what happens to dorsal cells if they lose their dorsal identity. Formerly, it was shown that these cells are eliminated from the dorsal compartment via apoptosis or through relocation to the ventral compartment. Here we show that a third mode of elimination, basal extrusion, also contributes to their clearing. We quantified, for the first time, contributions of each mechanism and discovered a regional bias in their operation. Importantly, if apoptosis is blocked, basal extrusion takes over to ensure clearance from all regions. Recent modeling

grants to F.H. (PP00P3_150682 and PP00P3_179075) and by the Ser Cymru II programme which is part-funded by Cardiff University and the European Regional Development Fund through the Welsh Government (80762-CU186). The funders had no role in study design, data collection and analysis, decision to publish, or preparation of the manuscript.

**Competing interests:** The authors have declared that no competing interests exist.

approaches suggested that there is a lower limit to the clone size for extrusion. Therefore, we tested the hypothesis that the choice of elimination mechanism may be dictated by the clone size. We prevented cell divisions within the clones to be eliminated and found that even 1–2 cell clones readily underwent basal extrusion, demonstrating that there is no lower limit to the clone size for extrusion.

## Introduction

Multicellularity requires precise spatial organization of cells during development. Aberrant cells that arise as a result of sporadic mutations or chromatin defects can challenge the robustness of a developmental program. For example, the cells that acquire incorrect positional identity disrupt the proper spatial organization. Those cells are potentially dangerous and cannot be tolerated. Therefore, mechanisms that ensure elimination of such cells are in place, yet they remain poorly understood. Arguably, one of the best-characterized systems to study the spatial organization of a developing tissue is the *Drosophila* wing imaginal disc. This organ is amenable to mosaic analysis techniques that are particularly useful for studying interactions of differently specified cells *in vivo*. The pattern in the wing disc is set by restricted expressions of fate determinants that are turned on in a sequential manner. Engrailed and Apterous (Ap) define posterior and dorsal fates, respectively, and lead to the formation of compartment boundaries that provide lineage restriction [1–3]. Further subdivisions are achieved by restricted expressions of Vestigial in the pouch, Homothorax and Teashirt in the hinge and the Iroquois complex in the notum [4–8]. The cell clones with altered expression of such genes disrupt the tissue pattern and trigger a set of common events. Such clones round up to minimize contact with their neighbors. Some of the clones were reported to undergo apoptosis or bulge out of the tissue forming cysts [9–14].

In addition to separating opposing compartments from each other and preventing cell mixing, the compartment boundaries also act as signaling centers. The morphogens Decapentaplegic (Dpp) and Wingless (Wg), secreted from these centers, form concentration gradients and orchestrate proper tissue size and shape [15–19]. Disruption of morphogen gradients also triggers a set of common events that will eventually restore the pattern via elimination of the mispositioned cells [9, 10, 13, 20–23]. Strikingly, in all these cases there is a strong regional bias. For example, *thickveins* mutant cell clones, that lack the Dpp receptor, undergo apoptosis or basal extrusion in the medial wing disc where the pathway activity is high, yet they can be tolerated laterally where the pathway activity is naturally low [21, 24, 25]. Therefore, disruption of the pattern in the tissue prompts the mechanisms in place to restore it.

Here, we aimed to understand how cells with altered expression of the dorsal determinant Ap are eliminated from the tissue. It has been shown that Ap expressing clones are eliminated from the ventral compartment; whereas clones expressing Ap inhibitor *Drosophila* LIM only (dLMO) are eliminated from the dorsal one [11]. Ap acts as a transcription factor [26]. Via regulation of its target genes glycosyltransferase Fringe and Notch ligand Serrate in dorsal compartment it mediates activation of Notch and Wg signaling pathways at the D/V compartment boundary [27–31]. Similar signaling was observed around Ap or dLMO cell clones if they happen to be in the incorrect place [11, 32]. Depletion of Notch signaling within the clones, using $Notch^{DN}$, was shown to prevent almost all elimination in the pouch region [11]. The same rescue effect was observed when the apoptosis of clones was prevented by expression of *p35* [11]. This highlights the importance of the ectopic signaling for the elimination process and defines apoptosis as its main executor.

Notably, all these experiments were focused on the clones located in the pouch. However, whether clones in other regions behave the same remains unknown. Here we take a quantitative approach to define the contributions of different strategies–apoptosis, basal extrusion and relocation–employed by the tissue to deal with mis-positioned cell clones. Importantly, we did not limit our analysis to a specific region, and characterized *ap* clone behavior throughout the tissue. Our approach revealed a striking regional bias of the contributing mechanisms.

# Results

## A new *apterous* allele and use of twin-spots allow systematic and quantitative analysis of clone elimination

Systematic analysis of the behavior of *ap* mutant clones has not been possible until recently due to technical limitations of classical clonal analysis approaches. This is because the *ap* locus lies between the centromere and the canonical flippase recognition target (FRT) site on the right arm of the second chromosome. Hence this FRT site cannot be used to generate *ap* mutant patches. To circumvent this problem, we used a new *ap* allele generated by Bieli et al. [33, 34], where a well-positioned FRT (f00878) was used to generate a deletion of the whole coding region of *ap (ap^{DG8})*. Using this new tool, we first generated positively marked *ap* mutant clones as well as clones with ectopic Ap expression. The wild-type control clones were distributed uniformly throughout the disc (Fig 1A). In contrast, clones altered for Ap function displayed compartment bias. In agreement with former reports [11, 32], cells that lose *ap* expression are underrepresented in the dorsal compartment (Fig 1B) and, likewise, cells with ectopic Ap expression are eliminated from the ventral compartment (Fig 1C). The misspecified clones that remained in the tissue minimize their contact with surrounding native cells suggested by their round shape and display ectopic boundary signaling (detected by Wg) at the clone border, where Ap-expressing and Ap- non-expressing cells contact each other (Fig 1B, arrows). Thus, cells are cleared from the region where they do not normally belong, pointing at the existence of intrinsic mechanisms that detect and get rid of misspecified cells and hence contribute to the maintenance of compartment organization.

In order to have a disc intrinsic measure of how many clones were originally generated we utilized a classical mitotic recombination approach that allows to generate mutant clones together with their wild-type twin sisters. To mark mutant clones positively we placed *GFP* on the chromosome that carried *ap* mutation. Therefore, in our set-up the *ap* homozygous mutant clones were marked positively by two copies of *GFP*, whereas their sister wild-type

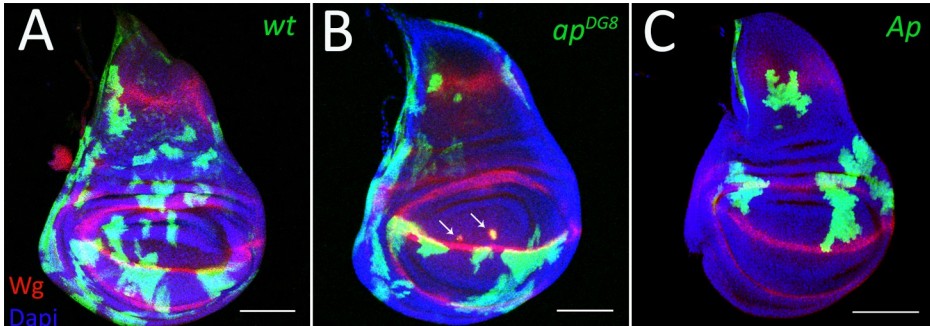

**Fig 1. The recovery of clones altered for Ap function is compartment biased.** (A-B) Third instar wing discs with GFP-marked wild-type (A), *ap^{DG8}* (B) and Ap-expressing (C) clones. The arrows point to ectopic Wg expression around mis-specified clones. Scale bars represent 100μm. Hereafter disc orientation is dorsal up, anterior to the left.

clones (twin spots)—by the absence of *GFP*. To understand what happens to the mutant cells after their induction, we followed their fate in a time-course experiment using this set-up. We induced clones shortly before D/V boundary formation, at 46h after egg laying (AEL), and took time points every 10h (Fig 2A). The dorsal clones of each time-point were categorized into 3 groups (Fig 2B). The first group includes pairs of *ap* mutant and wild-type clones (Fig 2B (a)). This group reflects the number of mutant clones that remained in the dorsal compartment at a particular time point. The second group contains wild-type clones without their mutant sisters (Fig 2B (b)) and corresponds to the number of *ap* mutant clones which had already been eliminated. The last group includes wild-type clones in the dorsal compartment that have their mutant twins in the ventral part (Fig 2B (c)), suggesting that those mutant clones are presumably clones of dorsal origin that had been relocated to the ventral compartment. The percentages of clone pairs in each group normalized to the number of all dorsally located wild-type clones are shown (Fig 2G–2I, red lines).

At 24h AHS (the first time-point), almost 75% of dorsally located wild-type clones had their mutant twins (Fig 2C and 2G). Interestingly, the clones and their sisters had similar sizes (about 8–12 cells) at that time point. This indicates that the mutant cells initially were able to grow in the dorsal compartment. However, 10 hours later (34h AHS) the amount of *ap* clones recovered in the dorsal part dropped sharply (Fig 2D and 2G). Less than 40% of mutant clones remained in the dorsal disc. Those clones were much smaller and more circular compared to their wild-type sisters. In the next 20 hours, the number of dorsally located mutant clones declined only slightly (Fig 2E, 2F and 2G). Nearly 30% of mutant clones had remained in the dorsal disc at 54h AHS (the last time point). Interestingly, the clones at the very proximal notum (disc tip) and lateral notum regions were not eliminated (Fig 2E, arrowheads).

The mutant clones that were removed from the dorsal compartment had been either relocated to the ventral one (Fig 2H) or eliminated from the disc tissue completely (Fig 2I). We observed relatively high number (15%) of dorsally located wild-type clones with their mutant sisters in the ventral compartment at the first time-point (70h AEL) (Fig 2C and 2H). The number of such clones nearly doubled at the second time-point (80h AEL) (Fig 2D and 2H). The relocated mutant cell clones accumulated at the D/V boundary from the ventral side (visible in Fig 2D–2F). Clone relocation is coupled to boundary reorganization (Fig 2C' and 2E', arrows). We observed that the ectopic signaling induced between the mutant cells and the surrounding Ap-expressing cells can be incorporated into the regular compartment boundary if a mutant clone happens to arise in close proximity to the D/V boundary. Importantly, the deformed D/V boundary straightens after the relocation has been completed, as we nearly never observed boundary deformations at 100h and later (Fig 2F').

The high percentage of relocation events (30% of all *ap* clones induced) raised the question of how many of these events were by chance, especially because the clones were induced prior to D/V boundary formation. To estimate the frequency of clones being born and twins ending in opposite compartments by chance, we analyzed control discs, where both sister clones were wild-type (Figs S1 and 2G–2I blue lines). In such control discs, nearly all dorsal clones (95%) remained in the dorsal compartment together with their twins (Fig 2G). The sister clones located in different compartments were observed very rarely (below 5%) (Fig 2H). Therefore, we conclude that dorsally originated *ap* mutant clones that are in close proximity to the D/V boundary are actively relocated into the ventral compartment.

Finally, we found that a high number of dorsal mutant clones (42%) were completely eliminated from the wing discs (Fig 2F and 2I). The majority of the elimination took place early, between the first two time points. Overall, 72% of all dorsal *ap* mutant clones were removed from the dorsal compartment: 30% *via* relocation and 42% *via* full elimination.

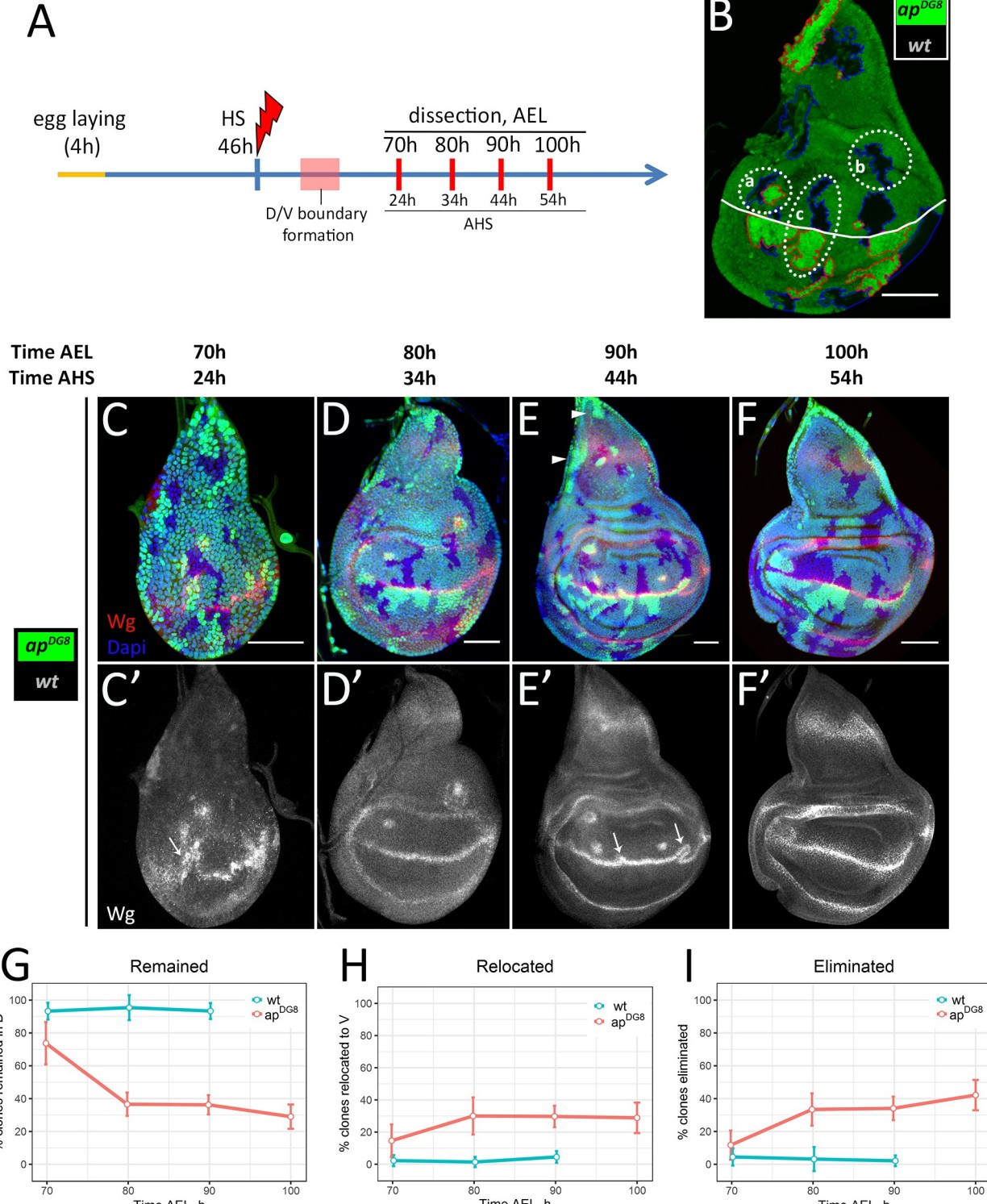

**Fig 2. The dynamics of clone elimination.** (A) Time-course scheme. (B) Strategy of clonal analysis. Example of the disc with *ap* mutant (2 copies of GFP, red outline) and wild-type sister (absence of GFP, blue outline) clones. (a)–wild-type clone together with the mutant twin; (b)–wild-type clone without the mutant sister; (c)–wild-type clone with *ap* mutant sister in the opposite compartment. White line corresponds to the D/V boundary. (C-F) Wing discs of indicated times containing differently marked wild-type and *ap^DG8* sister clones. (C'-F') Wg channel of C-F. (G-I) Plots represent amount of dorsal clones that remained in the dorsal part (G), have been relocated to the ventral part (H) or completely eliminated from the disc (I) as a function of time. Number of clones in each group was normalized to the number of dorsally located wild-type clones (per

disc). Red lines correspond to the ratios of $ap^{DG8}$ clones to their wt sisters (experimental discs); blue lines correspond to the ratios of wt clones to their wt sisters (control discs, shown on S1 Fig). Note, the control discs were analysed only at 70h, 80h and 90h AEL. A minimum of 15 discs were analysed for each time-point. Data represent mean±CI (95%). Scale bars represent 50μm.

## The later the clone induction, the less efficient is the elimination

Next we asked whether the elimination and relocation rates depend on the clone induction time. To address this question, we induced *ap* clones later, at 66h AEL (this timepoint is after boundary formation), and analyzed at 100h and 110h AEL, which correspond to 34h and 44h after heat-shock (AHS), respectively (Fig 3A). Thus, we can compare the results of this experiment (late induced clones) with the results of our previous experiment (early induced clones) at least for 34h and 44h AHS.

Expectedly, the clones were more abundant when induced in older discs due to the higher cell number (compare Fig 3B and 3C with Fig 2D and 2E). Using the categorization strategy described (Fig 2B), we quantified the percentage of mutant clones of dorsal origin that were either completely eliminated from the disc tissue or relocated to the ventral compartment. We found that the portions of completely eliminated mutant clones as well as relocated ones were significantly smaller for the late induced clones compared to the early induced ones at both 34h (Fig 3B and 3D) and 44h AHS (Fig 3C and 3E).

Remarkably, the induction of *ap* LOF clones after D/V boundary formation resulted in boundary deformations (Fig 3B', note the wiggly D/V boundary) similar to what we observed when the clones were induced before the D/V boundary formation (Fig 2E'). Therefore, the compartment boundary can be rearranged after its formation. Such boundary flexibility allows the mutant clones to be rescued by displacement to the ventral compartment, though this happens more rarely for the clones induced after boundary formation than for the ones induced before. Altogether, the efficiency of misspecified cell elimination depends on the developmental stage: the mutant clones induced early are eliminated from the dorsal compartment more efficiently than the late induced ones.

Previous studies showed that *ap* mutant clones can lead to deformations in the adult wings [3]. Indeed, in most cases, the presence of cells with inappropriate dorso-ventral positional identity in adult tissue caused ectopic margin formation (S2B Fig), wing margin duplications (S2C and S2C' Fig), blister-like outgrowths (S2D and S2E Fig), and, occasionally, wing duplications (S2F and S2G Fig). Importantly, the occurrence of defective wings highly correlates with the time of clone induction. When *ap* clones were induced late a vast majority of the wings (83%) had defects. In contrast, only one out of three wings were defective when the induction was early (Fig 3F). We observed the same trend with Ap-expressing clones induced at different times (Fig 3F). Thus, early induced misspecified clones are more likely to be eliminated, leading to normal wings. This finding highlights the importance of mechanisms in place to eliminate misspecified cells.

## Three region specific mechanisms ensure the clearance of misspecified cell clones

Next, we set out to study how the misspecified cells are being cleared from the tissue. As discussed, for *ap* mutant cells in close proximity of the D/V boundary, an elegant solution is to cross over to the opposite side (Fig 4A). This strategy also works for the clones misexpressing Ap (Fig 4B). During this process the ectopic boundary signaling induced between the misspecified cells and surrounding wild-type cells fuses with the regular D/V boundary, forming a loop-like structure around the misspecified clone (Fig 4A' and 4B'). This allows dorsal mutant

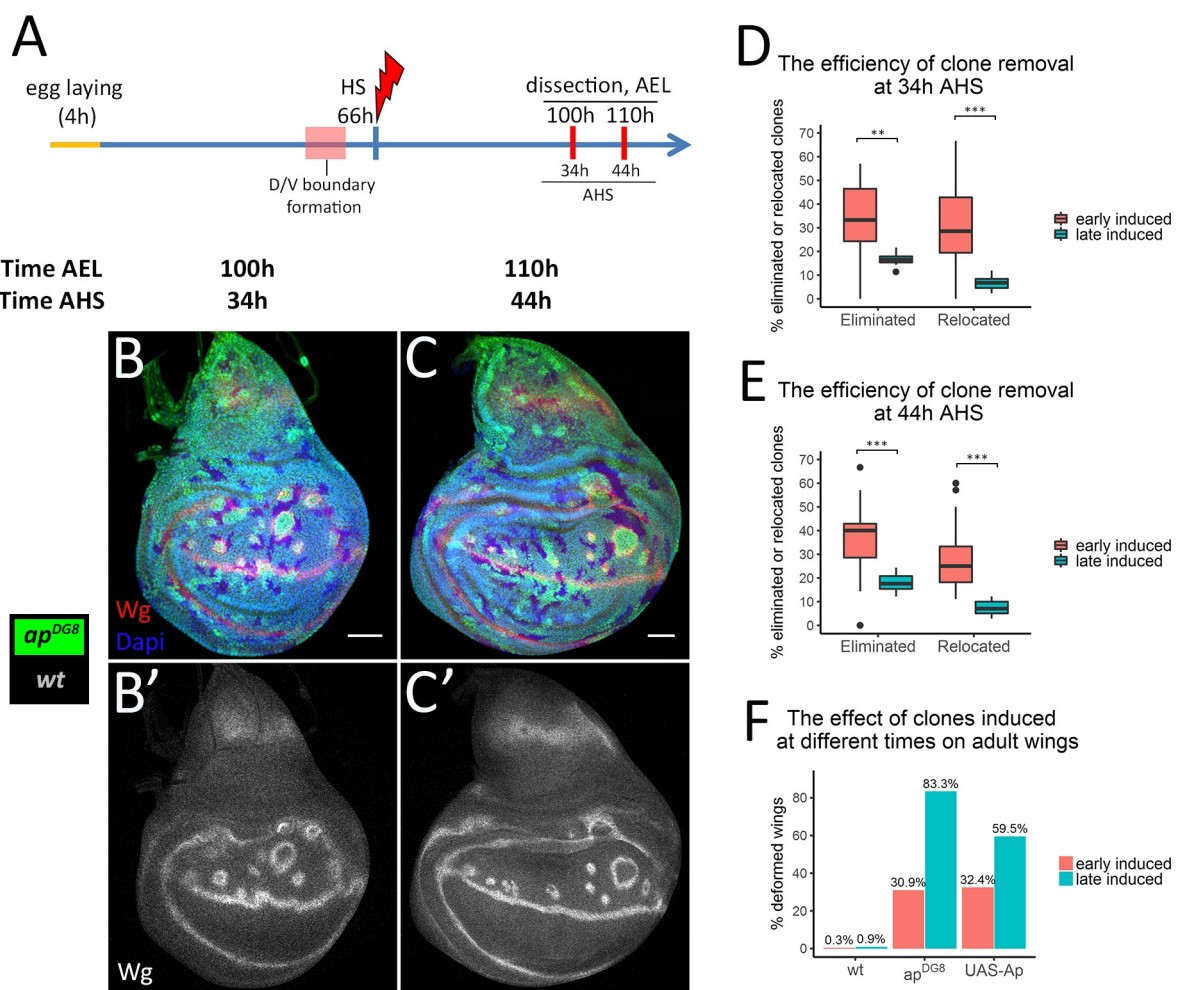

**Fig 3. The late induced clones are eliminated less efficiently than the early induced ones.** (A) Time-course scheme where clones are induced at 66h AEL. (B-C) Wing discs of indicated times containing differently marked $ap^{DG8}$ and wild-type clones. (B'-C') Wg channel alone. (D-E) Comparison of removal of $ap^{DG8}$ clones that were induced at 46h AEL ("early induced", shown in Fig 2) with the one of those that were induced at 66h AEL ("late induced") at 34h (D) and 44h (E) AHS. At least 10 discs with late induced clones were analysed per time point. (F) Percentages of defective wings due to wild-type, $ap^{DG8}$ and Ap-expressing flip-out clones induced at 46h or at 66h AEL. Numbers of analysed wings: early induced: wt—302; $ap^{DG8}$–304; UAS-Ap– 482; late induced: wt—230; $ap^{DG8}$–102; UAS-Ap– 84. Scale bars represent 50μm.

cells or ventral Ap-expressing cells to mix with the cells from the opposite compartment and eventually recover at the correct place.

Another mechanism contributing to the elimination of misspecified cells is apoptosis [11]. Indeed, as revealed by TUNEL assay, both *ap* mutant and Ap-expressing clones undergo apoptosis in the inappropriate compartment (Fig 4C–4D'). Interestingly, the apoptotic cells were detected both within and surrounding the misspecified cell clones (Fig 4C–4D', insets).

Moreover, some misspecified clones displayed evidence of basal extrusion. We define extrusion as strong morphological changes that lead to out of the epithelium displacement of any size clone, single cells to large cysts. The apical surfaces of extruding clones were narrower (Fig 4E), than their basal side (Fig 4E'), and the central cells were much shorter (Fig 4E, XZ and YZ). More lateral cells of the clone will eventually fuse above the gap forming a cyst-like structure with the apical side enclosed inside, contributing to the clearance.

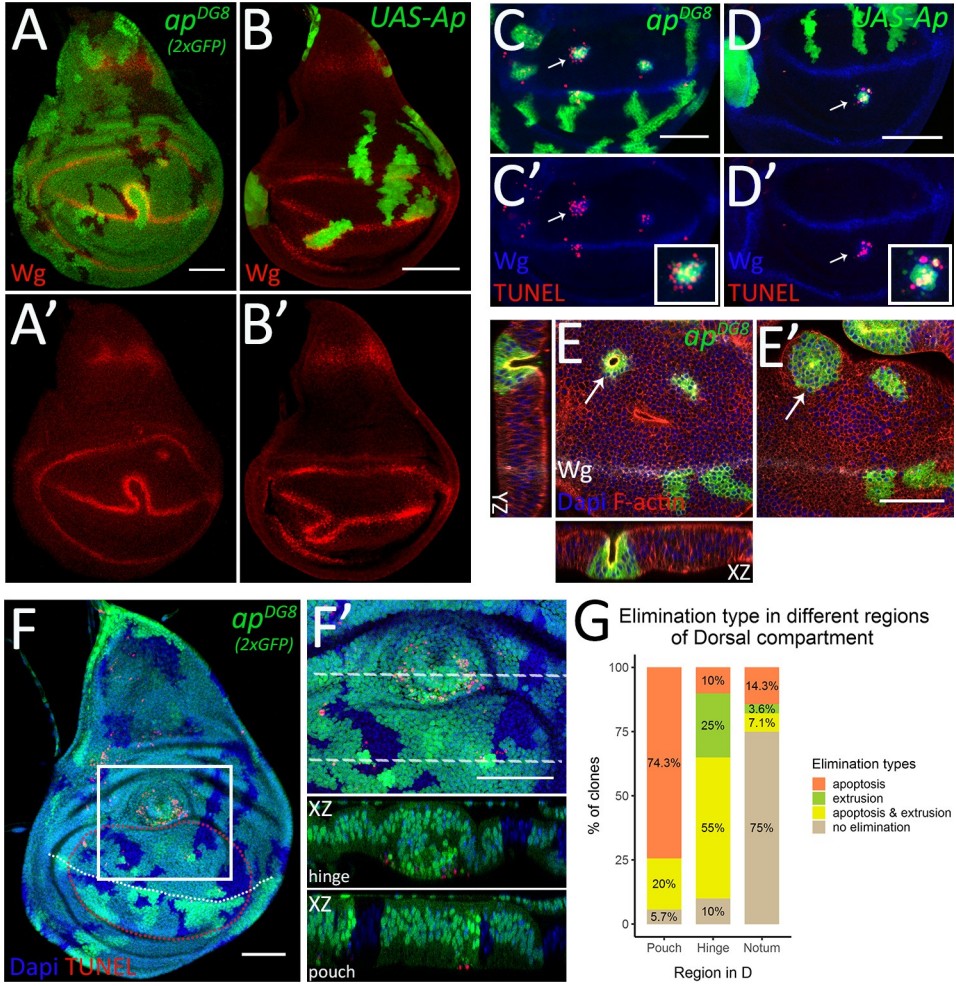

**Fig 4. Mechanisms of the elimination display region specificity.** (A-B) Third instar wing discs with $ap^{DG8}$ (A) or Ap-expressing (B) clones displaying D/V boundary deformation and clone relocation. (A'-B') Wg channel of A-B. (C-D) TUNEL assay of third instar wing discs with $ap^{DG8}$ MARCM (C) or Ap-expressing (D) clones. The pouch regions are shown. (C'-D') Wg and TUNEL channels of C-D. The insets show enlarged images of single representative clones defined by arrows. (E-E') Pouch region of the third instar wing disc with $ap^{DG8}$ clones shown from the apical (E) and the basal (E') sides. The arrows point to the bulging clone. The XZ and YZ planes throughout the bulging clone are also shown (XZ orientation: the apical side–up, YZ orientation: the apical side–right). (F-G) Mechanisms of elimination display region-specificity. (F) Third instar wing disc containing $ap^{DG8}$ mitotic clones (marked by 2 copies of GFP) that remained at 50h AHS. White dashed line represents D/V boundary; red dashed line outlines the pouch region (based on the most proximal fold). (F') Zoom-in of the region defined by white square in F. The XZ cross-sections throughout the clones located in the hinge and the pouch are shown below (orientation: the apical side–up). (G) Quantification of dorsal mutant clones in different regions depending on the evidence of elimination type: apoptosis, extrusion or apoptosis together with extrusion. A total of 83 dorsal $ap$ mutant clones from 23 discs were analyzed: 35 clones were in the pouch, 20 in the hinge and 28 in the notum. Scale bars represent 50μm.

Importantly, the vast majority of the bulging $ap$ mutant clones were in the prospective hinge or in the very proximal pouch regions of the dorsal compartment. To estimate if there is any relationship between the region of clone location and the mechanism of elimination we carefully analyzed all $ap$ mutant clones that remained in the dorsal compartment 50h after clone induction in the third instar wing discs. We scored the morphological changes indicating basal extrusion and the presence of apoptotic signal revealed by TUNEL assay. Mutant clones from 23 wing discs were analyzed. Clones located in different regions of the dorsal compartment (dorsal pouch, dorsal hinge and the notum) were grouped based on the type of

elimination they displayed: apoptosis (without evidence of extrusion), extrusion (without apoptosis), extrusion accompanied by apoptosis and the clones that did not display any evidence of elimination (Fig 4G). In the pouch (defined based on the most proximal fold), almost all misspecified clones underwent apoptosis (33 out of 35 clones) (Fig 4F and 4G). Some of these apoptotic clones also bulged out, especially the ones in the proximal pouch (7 out of 33). We found no examples of extrusion for the clones located near the D/V boundary. In contrast, in the hinge, the majority of the mutant clones displayed a cyst-like phenotype (16 out of 20) (Fig 4F and 4G). Interestingly, the bulging clones in the hinge were not necessarily accompanied by apoptosis, but all apoptotic clones displayed extrusion (Fig 4G). This finding suggests that the induction of clone extrusion in the hinge is not a consequence of apoptosis. The opposite scenario is more likely—apoptosis takes place following clone extrusion in the hinge. In the notum, 6 clones out of 28 examined contained apoptotic cells, and only 2 clones formed invaginations. The majority of remaining clones in the notum displayed evidence of neither apoptosis nor extrusion (Fig 4G). This analysis of clones remaining in the tissue 50h after clone induction revealed a clear regional bias in the choice of elimination mechanism. Next, to have a more complete picture of all elimination, we took into account the clones that were cleared off the dorsal compartment prior to our analysis. To have the numbers of those clones we counted wt twin spots that remained without a mutant sister or with a mutant sister in the opposite compartment (S3A Fig). We found that 50–70% of all mutant clones were already cleared from the dorsal region prior to our analysis. Relocation to the ventral compartment accounted for 31% of all clones in the dorsal pouch and 9% of those in the dorsal hinge, from the regions neighboring the D/V boundary. Notably, the majority of the twin-spots were without mutant sisters in the hinge and the notum, especially in its central part, suggesting that mutant clones were actively eliminated from this region earlier by either apoptosis or extrusion (S3A Fig).

Altogether these data demonstrate that misspecified cells are removed by three different mechanisms: relocation, apoptosis and basal extrusion. Moreover, apoptosis dominates in the pouch, whereas extrusion is the main mechanism in the hinge.

## Extrusion occurs independently of apoptosis and takes over in the absence of cell death

To assess the contribution of cell death to the elimination process, we prevented apoptosis in $ap^{DG8}$ cells by co-expression of the inhibitor of apoptosis p35. Wild-type, UAS-p35, $ap^{DG8}$ and $ap^{DG8}$ with p35 clones were induced at early second instar and the discs of mid-third instar larvae were analyzed. The clones expressing only p35 (Fig 5B) behaved similarly to the wild-type GFP-expressing clones (Fig 5A). In both cases, the clones did not display apoptosis, as revealed by the TUNEL assay. In contrast, dorsally located ap mutant clones induced apoptosis (Fig 5C). As expected, expression of p35 within the clones perfectly inhibited apoptosis of clonal cells, but did not prevent induction of apoptosis outside the clone (Fig 5D, upper insert). The expression of p35 in the mutant clones significantly increased the number of recovered clones (Fig 5D and 5E). However, the number of mutant clones expressing p35 was still lower than that of wild-type or p35-expressing clones (Fig 5E).

Thus, apoptosis inhibition only partially rescues the elimination of the misspecified clones. To determine whether apoptosis inhibition influenced the relocation efficiency and whether the clones are indeed eliminated from the tissue in the absence of apoptosis, we analyzed $ap^{DG8}$ clones expressing p35 together with their wild-type sisters. The clones were induced at 46h AEL and analyzed at 80h, 90h and 100h AEL (similar to our time-course experiment in Fig 2). The quantification of relocation events and comparison to the results obtained with the

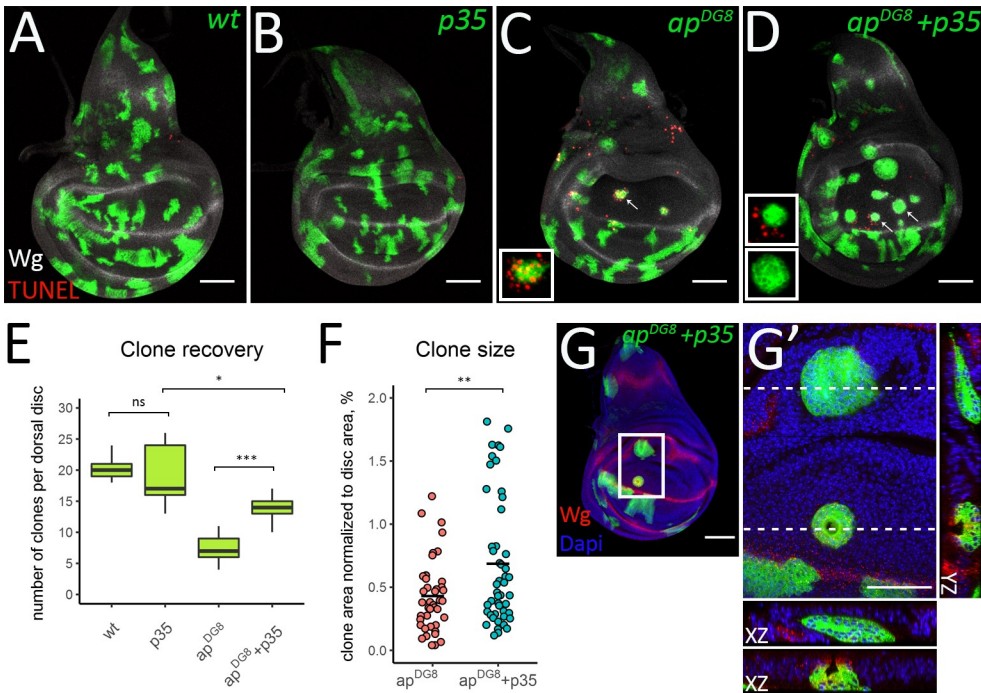

**Fig 5. In the absence of apoptosis misspecified clones become bigger and undergo extrusion.** (A-D) Third instar wing discs with wild-type (A), *p35*-expressing (B), *ap^DG8^* (C) and *ap^DG8^ p35*-expressing clones. The insets in C and D show enlarged images of single representative clones defined by arrows. (E) Clone recovery in the dorsal disc. 9 discs for each genotype were analyzed. (F) Plot shows areas of *ap^DG8^* and *ap^DG8^ p35*-expressing clones. (G-G') *ap* mutant clones are eliminated from the dorsal pouch via extrusion when apoptosis is blocked. (G) Third instar wing disc containing *ap^DG8^ p35*-expressing clones. (G') Zoom-in of the region defined by the white square in G. The XZ and YZ cross-sections throughout the clones located in the hinge and in the pouch are shown (XZ orientation: the apical side–up; YZ orientation: the apical side—left). Scale bars represent 50μm.

*ap* mutant clones alone (Fig 2), revealed that the expression of *p35* did not change the relocation efficiency at any time-point (S4A–S4D Fig). Approximately 30% of clones were relocated to the ventral compartment (S4D Fig). In contrast, the number of mutant clones that were fully eliminated from the disc tissue were significantly lower when apoptosis was blocked (S4E Fig). However, about 18% of dorsal wild-type clones were found without their mutant sisters (S4B and S4C Fig, arrows and S4E). This directly indicates that the misspecified clones can be eliminated from the tissue even in the absence of apoptosis. Many *ap* mutant clones with *p35* expression displayed evidence of basal extrusion. Interestingly, in this case cyst formation was observed not only in the hinge region but also in the notum and in the pouch (Fig 5G and 5G'). This suggests that extrusion does not depend on apoptosis and can serve as a back-up mechanism of clone elimination.

## Clone size is important for cyst-formation but not for clone elimination via extrusion

We wondered whether the size of the mutant clones or the tension they experience could contribute to the choice of elimination mode, as was observed by Bielmeier et al., and hence explain the regional bias [9]. To this end, we measured clone size and circularity as a proxy for the line tension at the clone border and plotted the data depending on the region (S3B Fig). The mutant clones were highly circular both in the pouch and in the hinge within a similar range, suggesting that differences in line tension alone cannot account for the regional bias in

elimination mechanism. The clones in the notum displayed a huge variability in terms of size and circularity. Notably, the central and peripheral notum clones acted distinctly. The clones in the central notum were similar in size and circularity to the hinge clones, while the peripheral notum clones were less circular and on average larger. These measurements suggest that there would be less pressure for elimination in the peripheral notum and accordingly, the *ap* mutant clones frequently remain there.

Apart from the wild-type like behavior of the peripheral notum clones, one significant difference was the size of the clones in the pouch. On average, *ap* mutant clones were smaller in the pouch compared to the other regions (S3B Fig). Therefore, we wondered whether clone size would be linked to the choice of elimination mechanism. This could also explain why clones in the pouch (and in the notum) begin extruding upon apoptosis inhibition: since many *ap* mutant clones in the pouch normally undergo apoptosis, they may not have a chance to reach the size required for extrusion, while apoptosis inhibition allows mutant clones to grow and reach a larger size (Fig 5F).

Thus, we asked whether changing the clone size affects their elimination in the presence and absence of apoptosis. To reduce the clone size we made use of *string (stg)* RNAi. Stg is an activator of the cyclin-dependent kinases. It regulates cell cycle progression by driving cells into mitosis [35]. Accordingly, *stgRNAi* expressing cells proliferate slowly and the clones have smaller size compared to the wild-type clones (S5A and S5C Fig). However and importantly the expression of *stgRNAi* did not affect the clone recovery rate (S5I Fig). Therefore, *stgRNAi* expression and associated with it reduction of proliferation do not cause clone elimination per se. As previously, *p35* was used to prevent apoptosis within the clones (S5B Fig). The clones expressing both *p35* and *stgRNAi* combined both effects: they were smaller, and were recovered at a higher rate than wild-type clones (S5D and S5I Fig). To modulate apoptosis and clone size in misspecified cells at the same time, we induced *dLMO* flip-out clones, which ectopically express dLMO and inhibit Ap function. Like *ap* mutant clones, *dLMO* flip-out clones induce ectopic boundary signaling and are efficiently eliminated from the dorsal compartment (Figs 6A, S5E and S5I; see also [11, 36]). The distribution of the elimination types the *dLMO* clones undergo in different regions mimics the one of *ap* mutant clones (Fig 6E, *dLMO*). Upon apoptosis inhibition the behavior of *dLMO* clones again resembled the behavior of *ap* mutant clones (Figs 6B and S5F): *p35* co-expression increased the clone recovery rate (S5I Fig) and led to clone extrusion in all regions of dorsal compartment (Fig 6E, *dLMO + p35*). Contrary to our expectations, the reduction of *dLMO* clone size did not influence the clone recovery rate (Figs 6C, S5G and S5I). These smaller *dLMO* clones were less likely to be associated with apoptosis and more frequently displayed invagination, shortening, and extrusion phenotypes, especially in the pouch and in the notum (Fig 6E, *dLMO + stgRNAi*). Moreover, when we reduce the size of *dLMO* expressing clones and prevent their apoptosis at the same time, most of misspecified clones underwent extrusion in all parts of the dorsal compartment (Fig 6D and 6E, *dLMO + stgRNAi + p35*). The fact that reduction of clone size does not prevent clone extrusion and does not increase the clone recovery rate suggest that elimination of misspecified clones via extrusion occurs regardless of the clone size.

Although the small size does not prevent the misspecified clones from being extruded, such clones extrude from the tissue in a different way than the larger ones do. Careful analysis of *dLMO + p35* and *dLMO + p35 + stgRNAi* clone morphology showed that *dLMO + p35* clones, which were generally of medium to large size (more than 6 cells), form cyst-like structures with a cavity inside, whereas small clones (1–6 cells) did not. Most clones expressing *stgRNAi* were clones of the small size. During our analysis, we found clones at different steps of extrusion process from which we could reconstruct the whole process for both the large (Fig 6F) and the small (Fig 6G) clones. At the first step the large clones experience shrinkage of the

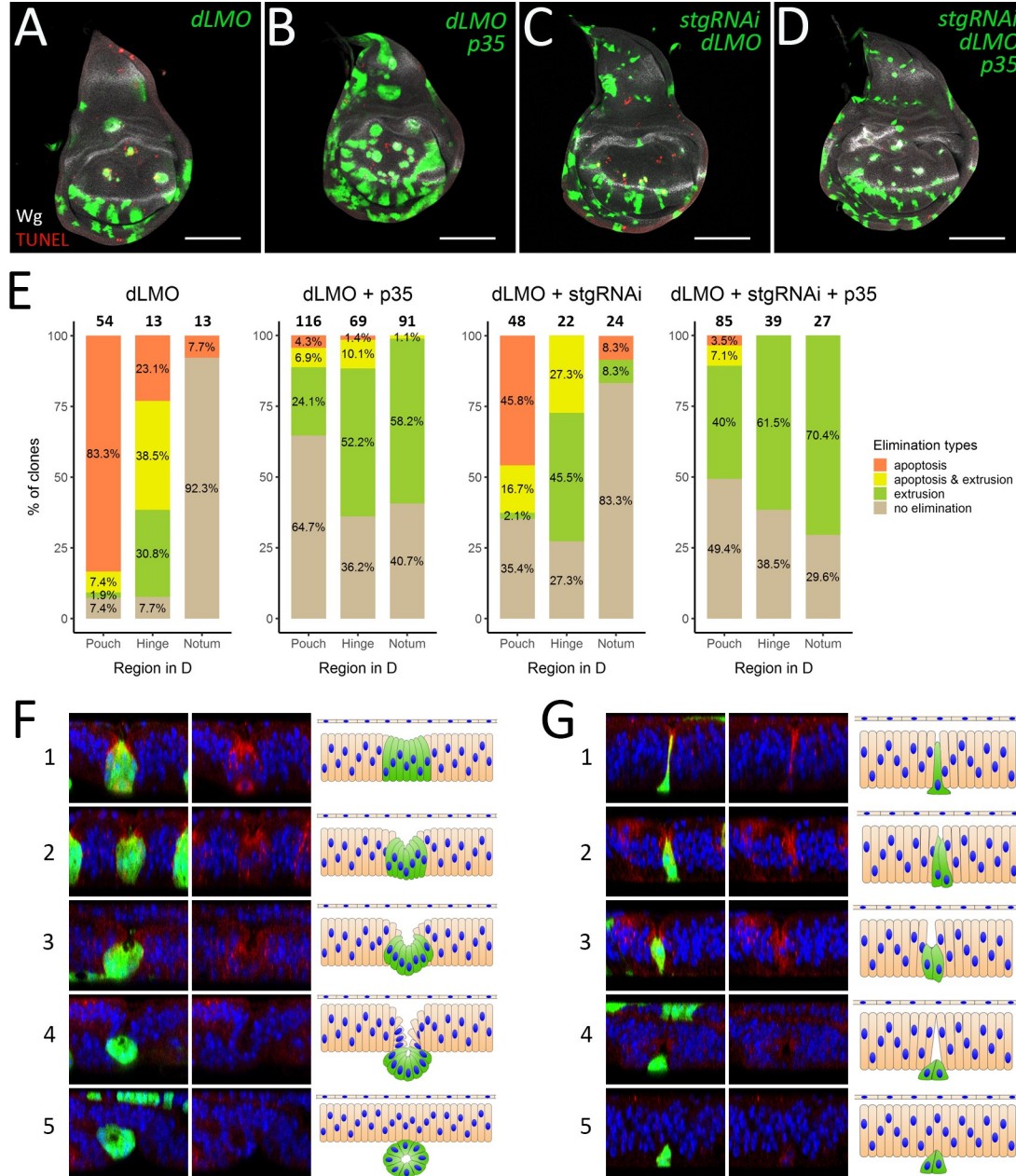

**Fig 6. Clone size reduction does not prevent clone extrusion.** (A-D) Third instar wing discs containing *dLMO* (A), *dLMO+p35* (B), *stgRNAi+dLMO* (C) and *stgRNAi+dLMO+p35* (D) clones. (E) Quantification of clones of the indicated genotypes in different regions of dorsal compartment depending on the evidence of elimination type: apoptosis, extrusion or apoptosis together with extrusion. The numbers of analyzed clones in each region are displayed above the bars. (F-G) Extrusion of large and small clones. Examples of large (*dLMO+p35*) (F) and small (*stgRNAi+dLMO+p35*) (G) misspecified clones at different stages (1–5) of extrusion process. XZ cross-sections throughout the clones are shown (orientation: the apical side–up). The left panel: clones in green, nuclear staining (Dapi) in blue, Wg staining in red; the middle panel—nuclear and Wg staining alone; right panel–schematic representation of morphological changes. The initial steps of extrusion of both large and small clones involve apical constriction and reduction of cell height (cell shortening) from the apical side (F and G, 1–2). Propagation of those processes, especially cell shortening, in case of large clones leads to cyst formation, where apical sides of clonal cells face the newly-formed cavity (F, 3–4). In contrast, small clones do not form cysts, although the cells reduce their height further and get extruded from the tissue (G, 3–4). Finally, the wild-type neighboring cells fuse above the clones and restore epithelial integrity (F and G, 5). Scale bars represent 100μm.

apical surface (Fig 6F-1). At the same time clonal cells, especially cells in the clone center, get shorter leading to clone invagination (Fig 6F-2). Eventually all cells in the clone are reduced in height and the clone forms a cyst-like structure (Fig 6F-3). The cyst is pushed out from the tissue plane and becomes enclosed (Fig 6F-4). After the cyst extrusion is complete, the disc epithelium restores its integrity (Fig 6F-5). The small clones also begin the extrusion process by constriction of their apical areas, expansion of the basal side and cell shortening (Fig 6G-1). These changes lead to local tissue invagination (Fig 6G-2 and 6G-3). Further reduction of clone height causes clone extrusion. At the same time, neighboring wild-type cells establish contacts above the extruding clone (Fig 6G-4) and the tissue restores its integrity and shape (Fig 6G-5). In conclusion, unlike large clones, small clones do not form cyst-like structures, however both types of clones can leave the tissue via basal extrusion. The apical construction and cell shortening are the common changes resulting in local tissue invagination and clone extrusion.

## Discussion

Here we studied the behavior of cells misspecified for the dorso-ventral identity. Using a non-canonical FRT site [33] we induced *ap* mutant clones and analyzed their behavior in the dorsal compartment. Interestingly, the misspecified cells are not eliminated immediately after induction. Initially, we suspected that the clones needed to reach a certain size to initiate elimination. However, our data shows that the clone size is not a decisive parameter for the elimination. The misspecified cell clones, as small as 1 cell, can be extruded from the epithelia. Alternatively, the effect might be due to Ap protein or the transcript stability. In this scenario, bringing Ap below a certain threshold simply requires time or several divisions that would dilute the protein level in each cell. Although the misspecified clones are able to grow within the first 24h, most of them are recognized and effectively eliminated from the dorsal compartment within the following 10h. We have defined 3 mechanisms that ensure their elimination: relocation to the opposite compartment, apoptosis and basal extrusion.

### Relocation to the opposite compartment

The phenomenon, when dorsal cells mutant for *ap* cross the boundary and join the ventral compartment, has been observed in the early work defining Ap as the dorsal determinant [32]. Importantly, this ability of cells to swap compartments according to their identity contributes to the elimination of misspecified cells. We found that up to 30% of mutant clones of dorsal origin leave the compartment via this mechanism. Three main events make clone relocation possible: the induction of ectopic boundary signaling around the clone, incorporation of this signaling into the compartment boundary, leading to loops protruding from the D/V boundary, and boundary straightening. How boundary straightening occurs is not known. However, it is very likely that the mechanisms that maintain the boundary straight during normal development are in effect here. For instance, it was shown that the D/V boundary has distinct physical parameters such as increased cell bond tension, cell elongation and oriented cell division, which tightly correlate with the boundary morphology and ensure its straightness [37]. Importantly, the increased tension depends on Ap and Notch activity [38]. Therefore, it is possible that the mechanical changes associated with displaced signaling help to bring D/V boundary to the normal shape. Notably, the ability of misspecified dorsal clones cross into the ventral compartment even after D/V boundary formation suggests that the signaling center is a very flexible and dynamic structure. It can be rearranged at any time during development in order to meet tissue needs.

## Apoptosis and extrusion

Although some misspecified clones, the ones that are close to the D/V boundary, can escape to the opposite compartment and survive, the majority of misspecified clones are completely eliminated from the disc tissue either via apoptosis or basal extrusion. Interestingly, apoptosis activation occurs in both the misspecified cells and the juxtaposing wild type cells. Moreover, inhibition of apoptosis in the clones does not prevent its non-autonomous activation. This suggests that apoptosis activation relies rather on interaction of cells with different fate identities than on misspecified cells themselves. Similar autonomous and non-autonomous activations of apoptosis was reported for the adjacent cells that experience discontinuity in the reception of either the Dpp or Wg signaling [21, 39].

A former study by Marco Milan and colleagues reported that *p35* co-expression rescued dLMO clones of dorsal origin completely [11]. In our set up the rescue effect of *p35* was also significant, however incomplete (Figs 5E and S4). We find that in addition to apoptosis, basal extrusion also contributes to the elimination of cells misspecified for the D/V position. The underlying reason for the discrepancy between the published results and ours might be the timing of clone induction, as the later induced clones are more likely to escape the elimination mechanisms in place. Another important factor that could contribute to the differences between *p35* rescue experiments is that the analysis in Milan paper was restricted to the pouch region, whereas we analyzed clones throughout the whole dorsal compartment. Therefore, quantification in the whole disc allowed us to recognize the contribution of basal extrusion to the process of clone elimination.

## Region specificity

Interestingly, apoptosis and extrusion display strong region preferences: apoptosis dominates in the pouch whereas extrusion occurs preferentially in the hinge. Extrusion as a mechanical process is determined by two parameters clone size and line tension along the clone boundary. Comparison of these physical parameters of pouch and hinge clones showed that although the pouch and hinge clones have similar circularity they significantly differ in size (S3B Fig). The hinge clones that are on average bigger than the clones in the pouch undergo extrusion more frequently compared to the pouch ones. In line with that, elegant work by Bielmeier et al. showed that the clones of intermediate size (40–80 cells) readily form cysts, whereas small clones (up to 20 cells) are less likely to buckle [9]. Thus, given similar line tension, differences in clone sizes could explain the regional bias of extrusion. However, our experiment in which we prevent clone growth and keep clones as small as several cells (by co-expression of *stg-RNAi*) clearly demonstrates that extrusion does not rely on clone size. Therefore, there is no lower limit to the clone size for extrusion, nonetheless, we do not exclude the possibility that clone size may play a role in defining the elimination mode. Apart from clone size, other factors by which the pouch and the hinge differ from each other can be involved in the determination of elimination strategy.

First, the cyto-architectural properties of the hinge and the pouch regions are different. Cells in the wing pouch have a long and narrow shape along their apical-basal axes, whereas cells in the hinge are shorter and wider [13, 40]. This makes the hinge region mechanically more disposed to bulging [41]. Second, the hinge region is resistant to irradiation and drug-induced apoptosis due to low levels of the pro-apoptotic gene *reaper* in that region [42]. Third, we find that *ap* mutant clones in the dorsal pouch and dorsal hinge have different effects on cell proliferation. The misspecified clones in the hinge increase cell proliferation in both an autonomous and a non-autonomous manner. By contrast, the clones in the pouch either grow at the normal rate or even slightly inhibit cell proliferation (S6 Fig). Most likely, ectopic

Notch/Wg signaling induced at the clone boundary mediates these effects. Indeed, it was reported that Notch or Wg misexpression increases cell proliferation and causes strong overgrowth in the hinge but not in the pouch [10, 13, 23, 43]. Thus, the extrusion of misspecified clones in the hinge could be driven by local crowding, which was shown to be linked to extrusion in the *Drosophila* pupal notum [44, 45]. However, our data suggests that the role of local crowding can be at most minor with regards to the extrusion of *ap* mutant clones. First of all, in the absence of apoptosis, the extrusion of *ap* clones occurs rather frequently not only in the hinge but also in the pouch and in the notum, where the clones do not induce overgrowth. In addition, the clones with artificially reduced size (*dLMO+p35+stgRNAi*) are still extruded from the epithelium despite the lack of the crowding effect (Fig 6E and 6G). Thus, the built-in apoptotic resistance in the hinge seems to be the most relevant determinant of elimination mode. However, a combination of several region specific factors that favors extrusion in the hinge and apoptosis in the pouch might be at work.

Here, we described three mechanisms that ensure clearance of cells with incorrect D/V identity and their regional bias. We also find that the elimination of misspecified cells is more efficient earlier in development. This suggests that the ability of developing tissue to remove inappropriately specified cells and actively maintain the compartment organization requires some tissue plasticity that diminishes over time.

## Materials and methods

### Fly stocks

The following fly stocks were used in this study: *ap^{DG8}* (described in Bieli et al. [33,34]), *FRT^{f00878}* (described in Bieli et al. [33,34]), *UAS-dLMO* (was kindly provided by Marco Milan), *UAS-Ap* (was kindly provided by Markus Affolter), *UAS-p35* [46]; *UAS-stgRNAi* (GD, 330033) obtained from the Vienna Drosophila Resource Center (VDRC). All crosses were kept on standard media at 25°C. Flipase expression was induced by a heat-shock at 37°C. The detailed fly genotypes and heat-shock induction conditions are presented in S1 Table.

### Immunostaining and sample preparation

Imaginal discs were prepared and stained using standard procedures. Briefly, larvae were dissected and fixed in 4% paraformaldehyde (PFA) in PBS for 20 min. Washes were performed in PBS + 0.03% Triton X-100 (PBT) and blocking in PBT+2% normal donkey serum (PBTN). Samples were incubated with primary antibodies overnight at 4°C. The primary antibodies used: mouse anti-Wg (1:2000, deposited to the DSHB by Cohen, S.M. (DSHB Hybridoma Product 4D4-concentrated)). Secondary antibodies were incubated for 2hr at room temperature. The secondary antibodies used: anti-mouse Alexa 568 and Alexa 633. Discs were mounted in Vectashield antifade mounting medium with Dapi (Vector Laboratories). For F-actin staining Phalloidin-Tetramethylrhodamine B (Fluka #77418) was added during incubation with secondary antibodies at the concentration 0.3 μM. For adult wing sample preparation, the flies of desired genotypes were collected and fixed in 70% ethanol. The wings were isolated and mounted in 3:1 Canada balsam: Methyl Salicylate.

### TUNEL assay

For the TUNEL assay In Situ Cell Death Detection kit, TMR red (Roche) was used. Larvae were dissected in cold PBS and fixed in 4% PFA for 1hr at 4°C. Samples were washed in PBT and blocked in PBTN for 1 hr. Next, samples were incubated with primary antibodies overnight at 4°C and with secondary antibodies for 4hr at 4°C. After washing the tissues were

blocked in PBTN overnight at 4˚C. Then, samples were permeabilized in 100 mM sodium citrate supplemented with 0.1% Triton X-100 and incubated in 50 μl of TUNEL reaction mix (prepared according to the recipe from the kit) for 2 hr at 37˚C in dark. After this step, the samples were washed in PBT for 30 min and mounted in Vectashield antifade mounting medium with Dapi (Vector Laboratories).

### EdU labeling

For the EdU assay Click-iT EdU Alexa Fluor 594 imaging kit (Invitrogen #C10339) was used. Larvae were dissected in Schneider's Medium at room temperature and incubated for 1 hr at 25˚C in 15 μM EdU working solution supplemented with 1% normal donkey serum. After the EdU incorporation, the tissue was washed in PBS supplemented with 3% bovine serum albumin (BSA) and fixed in 4% PFA for 20 min. Next steps, including blocking, incubation with primary and secondary antibodies, were done according to the standard immunostaining protocol. After washing the tissues were permeabilized by 3 washes (10 minutes each) in 0.1% Triton X-100 in PBS. The EdU reaction cocktail was prepared according to the recipe from the kit. The samples were incubated in 250 μl of the EdU reaction cocktail for 30 min at room temperature in dark. After that the samples were washed in PBT for 30 min and mounted in Vectashield antifade mounting medium with Dapi (Vector Laboratories).

### Image acquisition and analysis

**Image acquisition.** Image stacks of wing discs were acquired on Zeiss LSM710 or LSM880 confocal microscopes using 20x or 40x objectives. In most cases, 15–30 Z-sections, 1 μm apart, were collected. The images shown on Figs 4E, 4E', 4F', 5G', 6F and 6G and all images used for the analysis of the elimination type (Figs 4G and 6E) were acquired using a 40x objective. In this case, 80–130 Z-sections, 0.4–0.7 μm apart, were collected. Image stacks were projected using maximum projection in ImageJ.

**Definitions of regions.** Dorsal and ventral compartments were defined based on Wg staining. Pouch, hinge and notum were defined based on fold location: pouch–central region outlined by the first inner fold (Fig 4F, red dashed line); hinge–area between most inner and most outer folds (folded area); notum–area in the dorsal compartment above the outer fold.

**Elimination categories.** A clone was categorized as apoptotic if it contained at least 2 TUNEL positive cells. Extrusion was defined based on orthogonal views throughout clone centers; we looked for obvious invagination from the apical side. To ensure the invagination does not coincide with a fold, continuous invaginations beyond the clone were not counted. Clones that display both apoptotic and extrusion evidences were counted as a separate group ("apoptosis & extrusion"). Clones that displayed neither apoptosis nor extrusion at the moment of observation were counted as ("no elimination"). Eliminated / early eliminated mutant clones were determined based on wild type twin-spots that were not linked to any mutant clones. The location (pouch, hinge, notum) of those clones was defined based on the location of their wild type siblings. Relocated clones were counted as a number of mutant–wild type pairs where wild type clone was located in the dorsal part while mutant twin in the ventral one (Fig 2B(c)). We made sure that there were not any other attached clones (possible siblings) in the same compartment.

Clone area and circularity measurements were done on images preprocessed in ImageJ. Briefly, projected images (maximum projection) were filtered using Gaussian Blur (sigma 1.5) and manually thresholded to detect and select clones. The area and circularity of selected clones were measured using built in ImageJ functionality where circularity is defined as: $4\pi^*$ (area/perimeter$^2$).

Statistical analysis was done in R, v3.5.0. Conditions were compared using two-sample t-test. Comparisons with a p-value > 0.05 were marked as "ns" (non-significant); p-value ≤ 0.05 –"*"; p-value ≤ 0.01 –"**"; p-value ≤ 0.001 –"***".

## Supporting information

**S1 Fig. Relocation and elimination of wild type clones in the control discs are very rare events.** (A-C) Wing discs of the indicated times containing wild-type sister clones that are marked by either 2 copies of GFP or absence of GFP. (A'-C') Wg channel of A-C. Quantifications of the remained, relocated and eliminated wild-type clones are shown on the Fig 2G–2I, blue lines. Scale bars represent 50μm.
(TIF)

**S2 Fig. *ap* mutant cell clones cause deformations in adult wings.** (A) Wild-type wing. (B-G) Wings after induction of $ap^{DG8}$ clones during second instar contain different deformations: ectopic margin formation (B); wing margin duplication (C-C', arrowheads); blister-like outgrowths (D-E); and wing duplications (F-G). Scale bars represent 500μm.
(TIF)

**S3 Fig. Clone size, circularity and elimination mode depending on the region.** (A) Quantification of dorsal mutant clones in different regions depending on the evidence of elimination type including clones that have been eliminated (early eliminated) or relocated by the time of analysis. The analysis was based on quantification of wt clones and their mutant sister clones. A total of 213 dorsal wt clones from 23 discs were analyzed: 87 clones were in the pouch, 67 in the hinge and 59 in the notum. (B) The circularity and size of mutant clones remaining in different regions of the dorsal compartment at 50h AHS. The clones in the notum were grouped into 2 categories depending on whether they touch the sides of the disc (notum_Periph) or are in the central part (notum_Centr). Altogether 91 clones were measured.
(TIF)

**S4 Fig. Apoptosis inhibition does not rescue all mis-specified clones.** (A-C) Wing imaginal discs of indicated times with $ap^{DG8}$ clones expressing *p35* (marked by two copies of GFP) and wild-type sister clones (marked by the absence of GFP). Arrows point to wild-type clones that lost their mutant sisters; (D-E) Comparison of the amount of $ap^{DG8}$ clones (data from the Fig 2) with the amount of $ap^{DG8}$ + *p35* clones that were relocated to the ventral compartment (D) or completely eliminated (E). At least 15 discs with $ap^{DG8}$ clones and 12 discs with $ap^{DG8}$ + *p35* clones were analyzed. Scale bars represent 50μm.
(TIF)

**S5 Fig. The reduction of clone size does not affect their recovery.** (A-H) Third instar wing discs containing wild-type (A), *p35* (B), *stgRNAi* (C), p35+stgRNAi (D), *dLMO* (E), *dLMO* +*p35* (F), *stgRNAi+dLMO* (G) and *stgRNAi+dLMO+p35* (H) clones. (I) Clone recovery rate in dorsal compartment for each genotype. Scale bars represent 100μm.
(TIF)

**S6 Fig. *ap* mutant clones increase cell proliferation in the dorsal hinge but not in the dorsal pouch.** EdU cell proliferation assay of the third instar wing disc containing $ap^{DG8}$ clones. (A) Merged image ($ap^{DG8}$ clones, EdU and Wg staining). (A') EdU channel alone. (A") EdU and Wg channels. (A''') $ap^{DG8}$ clones and EdU staining. The insets show enlarged images of single clones from dorsal pouch (P) and dorsal hinge (H). Scale bar represents 50μm.
(TIF)

**S1 Table. Genotypes and experimental conditions.** Detailed genotypes and experimental conditions of data represented in the figures.
(DOCX)

## Acknowledgments

We are grateful to M. Müller, D. Bieli and M. Affolter of Biozentrum Basel for reagents and discussions. We thank M. Milan (IRB, Barcelona) and VDRC for providing fly stocks and DSHB for anti-Wg antibodies. Confocal microscopy was performed at the Cellular Imaging Facility of the University of Lausanne. We thank V. Dion (Dementia Research Institute, Cardiff University) and C. Hogan (European Cancer Stem Cell Research Institute, Cardiff University) for comments on the manuscript.

## Author Contributions

**Conceptualization:** Olga Klipa, Fisun Hamaratoglu.

**Data curation:** Olga Klipa.

**Formal analysis:** Olga Klipa.

**Funding acquisition:** Fisun Hamaratoglu.

**Investigation:** Olga Klipa.

**Methodology:** Olga Klipa, Fisun Hamaratoglu.

**Project administration:** Fisun Hamaratoglu.

**Supervision:** Fisun Hamaratoglu.

**Validation:** Olga Klipa.

**Visualization:** Olga Klipa.

**Writing – original draft:** Olga Klipa.

**Writing – review & editing:** Fisun Hamaratoglu.

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
