## [Decision Letter · Decision Letter 0]

6 Dec 2019

Dear Fisun,

Thank you very much for submitting your Research Article entitled 'Cell elimination strategies upon identity switch via modulation of apterous in Drosophila wing disc' to PLOS Genetics. Your manuscript, transferred and revsied from PLoS Biology, was evaluated at the editorial level and has been seen by one of the four initial reviewers. This reviewer supported publication but identified some minor aspects of the manuscript that should be improved.

We therefore ask you to modify the manuscript according to the review recommendations before we can consider your manuscript for acceptance. Your revisions should address the specific points made by this reviewer.

[LINK]

Yours sincerely,

François Schweisguth

Guest Editor

PLOS Genetics

Gregory P. Copenhaver

Editor-in-Chief

PLOS Genetics

Reviewer's Responses to Questions

**Comments to the Authors:**

Reviewer #1: The manuscript has improved and povides now interesting complementary quantifications. The manuscript is well written, the message is clear and supported by very carefull experiments. It illustrates the plasticity of the elimination processes used to get rid of misspecified cells. At this stage, I will therefore support publication. I only have few minor comments that may lead to text editing:

- line 381 "we wonder whether the size of the mutant clones or the tension...", I guess the authors should quote here the paper from Beilmeier and colleagues (Curr Biol 2016), otherwhise it is difficult to see where the rational comes from.

- line 424-427: " reduction of clone size does not increase the clone recovery rate" : actually I would have expected the contrary (reduction of clone size leading to a reduction of recovery rate), with a constant rate of apoptosis, if proliferation is reduced one would expect more clone disappearance. It is very interesting that the authors don't observe that. Maybe the authors should explain why they were expecting an increase of the recovery rate upon clone size reduction ?

- Discussion line 510-511: When refering to apoptosis occuring in and outside the clones, maybe the authors could quote the work from Piddini E and Vincent JP (Dev Cell 2011) showing that activation of Wg in clones can trigger apoptosis in the neighbouring cells.

- Methods: It would be good to give a bit more of details on the quantifications (measurement of clone area, determination of the location of the sibbling clones and of the category of elimination...)

**Have all data underlying the figures and results presented in the manuscript been provided?**

Reviewer #1: Yes

PLOS authors have the option to publish the peer review history of their article (what does this mean?). If published, this will include your full peer review and any attached files.

Reviewer #1: No

---

## [Editor Report · Decision Letter 1]

17 Dec 2019

Dear Dr Fisun Hamaratoglu,

We are pleased to inform you that your manuscript entitled "Cell elimination strategies upon identity switch via modulation of apterous in Drosophila wing disc" has been editorially accepted for publication in PLOS Genetics. Congratulations!

Yours sincerely,

François Schweisguth

Guest Editor

PLOS Genetics

Gregory P. Copenhaver

Editor-in-Chief

PLOS Genetics

Comments from the reviewers (if applicable):

**Data Deposition**

http://datadryad.org/submit?journalID=pgenetics&manu=PGENETICS-D-19-01748R1

Press Queries

---

## [Editor Report · Acceptance letter]

20 Dec 2019

PGENETICS-D-19-01748R1 

Cell elimination strategies upon identity switch via modulation of *apterous* in *Drosophila* wing disc 

Dear Dr Hamaratoglu, 

We are pleased to inform you that your manuscript entitled "Cell elimination strategies upon identity switch via modulation of *apterous* in *Drosophila* wing disc" has been formally accepted for publication in PLOS Genetics! Your manuscript is now with our production department and you will be notified of the publication date in due course.

With kind regards,

Matt Lyles

PLOS Genetics

On behalf of:
